# Ectoparasite survey of schoolchildren in the Republic of Guinea

**Elisabeth Yawa Diawara**[1,2,3], **Alpha Kabinet Keita**[4,5], **Alissa Hammoud**[2], **Adama Zan Diarra**[2,3,6], **Basma Ouarti**[2], **Coralie L'Ollivier**[2,3], **Stéphane Ranque**[2,3]*

**1** Institut National de Santé Publique (INSP), Conakry, République de Guinée, **2** Institut Hospitalier Universitaire (IHU) Méditerranée Infection Marseille, Marseille, France, **3** Aix Marseille Université, Service de Santé des Armées, RITMES, Marseille, France, **4** Centre de Recherche et de Formation en Infectiologie de Guinée (CRFIG), Conakry, République de Guinée, **5** Université Gamal Abdel Nasser de Conakry (UGANG), Conakry, République de Guinée, **6** IRD, MINES, Marseille, France

* stephane.ranque@univ-amu.fr

## Abstract

Ectoparasitoses, caused by infestations of highly contagious and ubiquitous parasites, include *Pediculus humanus* capitis (the head louse) and human scabies. These diseases affect many school-age children and are a major public health problem in low-income countries such as the Republic of Guinea. These parasites, such as lice, are known to be potential disease vectors. Given the lack of data on these conditions in Guinea, we conducted a nationwide epidemiological study to assess their prevalence in schools. In March 2023, 81 head lice were collected from 18 schoolgirls in the regions of Boké and Kankan. Molecular analyses were performed on all samples to assess genetic diversity and screen for pathogens. Using the A-D and B-C/E clade system applied to all 79 samples, it was found that 77 (97%) belonged to clade E, while two (3%) belonged to clade A. Three haplotypes were identified from clade E, including a new haplotype not previously described. The search for louse-borne pathogens in 81 samples revealed the presence of Acinetobacter spp. DNA in 37 (46%) of the lice collected. Specific qPCR analysis revealed the presence of *Acinetobacter baumannii* in three (8%) of the samples. Clinical lesions consistent with scabies were found in 20 children aged 3–17 years, 11 in Boké, 7 in Guéckédou, 1 in Kankan and 1 in Labé. Severe pruritus was reported in 40% of cases, mild pruritus in 30% and moderate pruritus in 25%.

## 1. Introduction

Ectoparasite infestation is a major public health problem affecting a significant number of school-aged children in low-income countries [1]. The mite that causes scabies and the lice that cause pediculosis are common parasites of the human skin, causing irritation and discomfort [2,3]. Scabies and pediculosis are among the oldest and most common parasitic diseases of humans [4]. These parasites are highly host specific and represent

**Data availability statement:** The raw data can be accessed at DOI: 10.13140/RG.2.2.16228.69769 Other data are available in the Supporting information file.

**Funding:** EYD was funded by the Fondation Méditerranée Infection. The funders had no role in study design, data collection and analysis, decision to publish, or preparation of the manuscript.

**Competing interests:** The authors have declared that no competing interests exist.

a significant public health problem worldwide, irrespective of economic and cultural development levels [5]. The primary mode of transmission is direct. The epidemiology of these diseases is influenced by a variety of socio-demographic and economic factors.

Lice are wingless insects of the order Phthiraptera, found mainly on mammals and birds. They lack the ability to fly and are permanently attached to their human hosts [6]. There are approximately 4,900 species of lice, with each species being specific to its host. These species are divided into five suborders: Anoplura, Amblycera, Ischnocera, and Rhynchophthirina. Anoplura, or sucking lice, are exclusively associated with mammals [7], while other suborders, such as Amblycera (or Mallophaga) chewing lice, are found on scales, lymph, and other matter [8]. Chewing lice are mainly found on birds and secondary mammals, and they feed mainly on feathers, blood, dead skin and secretions of their hosts [7]. Of these, only three species from the suborder Anoplura have been identified as human parasites: the head louse (*Pediculus humanus capitis*), the body louse (*Pediculus humanus*), and the pubic louse (*Pthirus pubis*) [8].

Body lice are known to be vectors for the transmission of important human pathogens such as *Rickettsia prowazekii*, which causes epidemic typhus, *Bartonella quintana*, which causes trench fever, and *Borrelia recurrentis*, which causes relapsing fever. Some authors have suggested that body lice may play a role in the transmission of *Yersinia pestis*, the causative agent of plague [9]. Furthermore, the role of head lice as vectors of pathogens has been demonstrated in a recent study, in which the presence of live *B. quintana* was observed in head lice, particularly during a localised trench fever outbreak in Senegal [10].

The human parasite Sarcoptes *scabiei* var*. hominis* causes scabies, a contagious and itchy disease transmitted by person-to-person contact. Although present in all countries, scabies is particularly prevalent in many tropical and poverty-stricken regions, affecting mainly children and the elderly. The prevalence of scabies in children varies from 5% to 50% in these areas. In 2017, the World Health Organization (WHO) classified it as a neglected tropical disease affecting more than 200 million people worldwide [8,11]. Scabies-causing mites can survive outside their host for up to four days, increasing the risk of reinfestation. Before dying, the female mite burrows under the skin to lay 10–25 eggs. These eggs typically hatch within three days, and the resulting larvae migrate to the surface of the skin where they mature into adults within 14–17 days [12]. Scabies is a significant health problem in Africa, particularly among children, with high prevalence reported in various settings including prisons, pre-schools and communities [13,14]. However, control of the disease is hampered by a number of barriers, including a shortage of specialists and a lack of diagnostic capacity in health facilities [15–17]. People of low socioeconomic status, often living in rural areas with limited access to health care, are most affected [18].

In Guinea, pediculosis and scabies have received little attention and are the subject of few studies. These diseases are common in schools, where hygiene is poor and children are often overcrowded but data on these diseases are scarce and come mainly from hospital studies, which are not representative of the general population. Therefore, we conducted a nationwide epidemiological study to assess the prevalence of ectoparasites in schoolchildren.

## 2. Patients and methods

### 2.1. Study design

We conducted a descriptive cross-sectional study of primary school children aged 3–17 years in 8 schools located in urban or rural areas of Guinea during the dry season of March 2023. We assessed the presence of pediculosis and scabies in these 8 primary schools located in four regions, namely: i) in the Boké region: Goreye (10.93N - 14.2930W) in the town of Boké and Tassara (10.7725 N - 14.3811W) in the sub-prefecture of Kolaboui; ii) in the Labé region: Kouroula (11,3147 N - 12,2858W) in the city of Labé and Garambe (11,2783N - 12,3416W) in the sub-prefecture of Garambe; iii) in the region of Kankan Bordo 1 (10,3825N-9,3305W) in the city of Kankan and Diankana (10,4761N-9,2327W) in the sub-prefecture of Karifamoriah; and (iv) in the region of Nzérékoré: Gbangbaïssa (8,5552N-10,1155W) in the town of Guéckedou and Télékoulo Centre (8,5461N-9,9516W) in the sub-prefecture of Télékoulo. These study sites are located on the country map in (Fig 1). The general sanitary and hygienic conditions were poor.

### 2.2. Study population inclusion criteria

Children were enrolled following a block randomisation per class in each school, in short, sampling was proportional to the number of pupils in each class, regardless of their status or relation to the pathologies of interest. Demographic, clinical

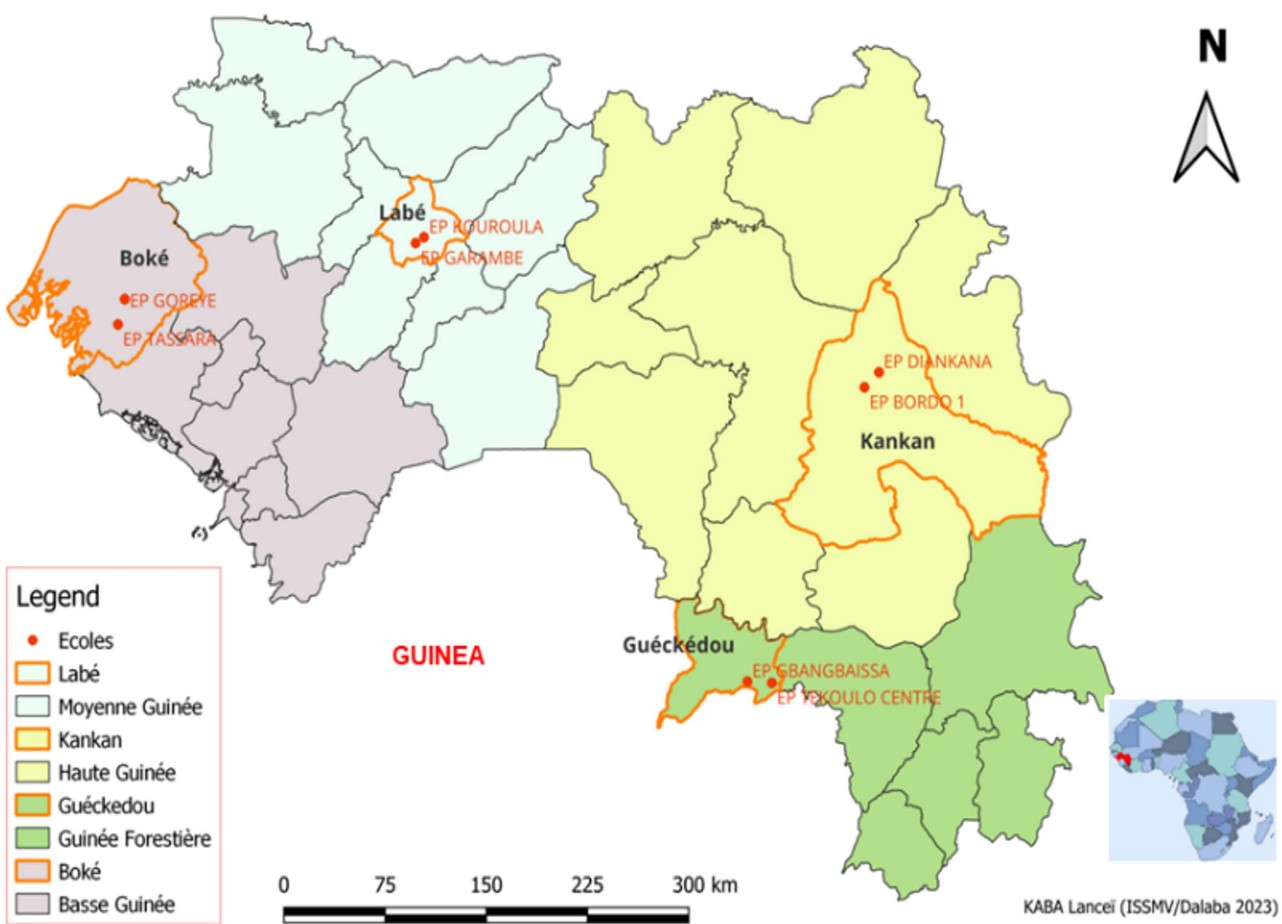

**Fig 1. Study sites in the Republic of Guinea.** (Base map and data from OpenStreetMap and OpenStreetMap Foundation.).

and risk factor data were collected using standardised questionnaires with the KoboToolbox data collection tool (https://www.kobotoolbox.org). For head lice, the presence of mobile lice, nits, cervical lymphadenopathy and duration of infestation were recorded. The lice were extracted using a fine-toothed comb. For suspected scabies, the presence of ridges, pustules, pruritus, the degree of itching and the presence of similar lesions in the child's environment were recorded, and suspected scabies lesions were swabbed with a sterile swab.

## 2.3. Ethical and regulatory issues

All randomised primary school children who had informed consent from their parents or guardians were included in the study. Only those with symptoms of head lice and/or scabies were sampled. Children who refused to participate in the study (themselves, their parents or legal representatives) were excluded. The study protocol was approved by the National Health Research Ethics Committee of the Republic of Guinea (No. 041/CNERS/23). The education managers, school directors and village elders accompanied the researchers throughout the study.

## 2.4. Samples collection and storage

All enrolled children were thoroughly examined for the presence of head lice and scabies. Lice were collected on site from infested children. Clinical lesions suspected of being scabies were observed by the principal investigator using a dermatoscope, and skin samples were taken from suspected patients for molecular diagnosis of scabies. Samples were taken from all identified lesions. The collected lice were preserved in 70% alcohol, while the scabies lesion samples were stored at +4°C. The samples were then sent to the Parasitology and Mycology Laboratory of the IHU Méditerranée Infection in Marseille (# ER36–2023).

## 2.5. Lice pre-treatment and morphological identification

In May 2023, eighty-one lice that had been preserved in 70% alcohol since March were pre-treated before analysis. The lice were first rinsed with distilled water and then dried on blotting paper. Their characteristics were carefully observed with a ZEISS Axio Zoom V16 binocular loupe (France) and documented by photographs taken with a Canon EOS 7D digital camera (France) equipped with a Canon MP-E 65 mm lens. Morphological keys provided by François-Xavier Pajot [19] were used for morphological identification prior to dissection of the lice for MALDI-TOF and DNA analysis.

## 2.6. MALDI-TOF MS analysis of the lice

**2.6.1. Pretreatment.** Lice preserved in 70% alcohol were pre-treated before analysis. The samples were superficially decontaminated in 70% ethanol, rinsed in sterile distilled water for 2 min and dried at room temperature for 5 min. Their characteristics were carefully observed with a ZEISS Axio Zoom V16 binocular loupe (France) and documented by photographs taken with a Canon EOS 7D digital camera (France) equipped with a Canon MP-E 65 mm lens (Fig 2). After cleaning and drying, the lice were dissected with a sterile scalpel. Cephalothoraxes were reserved for MALDI-TOF MS analysis [20], while abdomens were frozen at -20°C for subsequent PCR assays (Fig 3). Cephalothorax were then incubated in a dry heat bath at 37°C overnight to remove residual organic solvents [21]. The samples were then homogenised using a TissueLyser (Qiagen) with a pinch of glass powder (Sigma, Lyon, France) in a homogenisation buffer consisting of a mixture (50/50) of 70% (v/v) formic acid (Sigma-Aldrich, Lyon, France) and 50% (v/v) acetonitrile (Flüka, Buchs, Switzerland) for protein extraction as previously described [20].

**2.6.2. Sample loading on MALDI-TOF target plates.** After homogenisation, samples were centrifuged at 200 g for 1 min for debris removal. Next, 1 µL of supernatant was quadruplicated onto a MALDI-TOF SMtarget plate (Bruker Daltonics, Wissembourg, France) and coated with 1 µL CHCA matrix solution [saturated alpha-cyano-4-hydroxycinnamic acid (Sigma, Lyon, France), 50% acetonitrile (v/v), 2.5% trifluoroacetic acid (v/v) (Aldrich, Dorset, UK) and HPLC grade water]. The target sample was dried for a few minutes at room temperature. It was then loaded into a MALDI-TOF

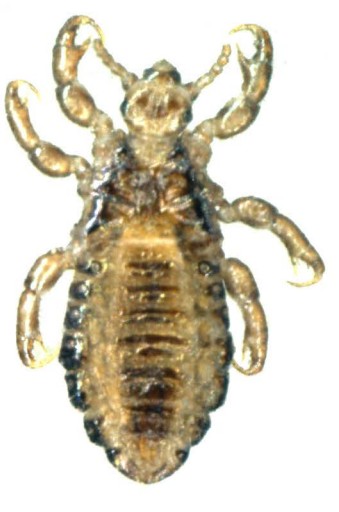
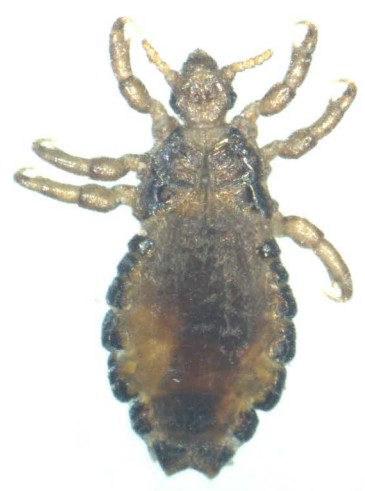

Male

Female

**Fig 2. Morphology of male and female adult lice (Canon EOS 7D digital camera photography).**

Microflex LT mass spectrometer (Bruker Daltonics) for analysis. The negative quality control was the matrix solution alone. The positive control was *Aedex albopictus* prepared under the same conditions.

**2.6.3. MALDI-TOF MS parameters.** Protein spectra were generated using a MALDI-TOF Microflex[LT] mass, spectrometer (Bruker Daltonics, Germany) with positive ion linear mode detection, operating at a laser frequency of 50 Hz and covering a mass range of 2–20 kDa. The acceleration voltage was 20 kV and the extraction delay was 200 ns. Each spectrum consisted of 240 laser shots in six regions of the same spot and was acquired automatically using the Auto Xecute method in the flex Control v2.4 software (Bruker Daltonics). Spectra were analysed using flex Analysis v3.3 software and then exported to ClinPro Tools v2.2 and MALDI-Biotyper v3.0 (Bruker Daltonics, Germany) for data processing (smoothing) [20].

**2.6.4. MALDI-TOF MS spectra analysis.** MS spectra were visually inspected using flexAnalysis v3.3 software (Bruker Daltonics, Bremen, Germany). The MS spectra were then transferred to ClinProTools v2.2 and MALDI-Biotyper v3.0 (Bruker Daltonics, Bremen, Germany) for data analysis, including smoothing, baseline subtraction and peak selection. Reproducibility of MS spectra was assessed by comparing the main spectral profiles (MSP) obtained from the four spots for each sample using MALDI-Biotyper v3.0 software (Bruker Daltonics, Bremen, Germany). Lice with a good score in our study were used to perform the blinding test.

**2.6.5. Reference spectra database construction.** Reference MS spectra were generated from the louse cephalothorax spectra using MALDI-Biotyper v3.0 software (Bruker Daltonics, Bremen, Germany) [22], using an unbiased algorithm that incorporates peak position, intensity, and frequency information.

**2.6.6. Lice identification blind test.** As there is no specific database for head lice, five samples with good results were selected as the baseline for the blind test. The reliability of species identification was assessed using the MALDI Biotyper v3.0 log-score values, which ranged from 1,708–2,389. Identification was considered reliable if the log-score value was greater than or equal to 1.7.

## 2.7. DNA-based analyses

**2.7.1. DNA extraction.** The remaining half of the dissected lice were pre-treated by adding 20 μL of Proteinase K solution to a collection tube containing 200 μL of G2 buffer, and the mixture was incubated overnight at 56°C prior to

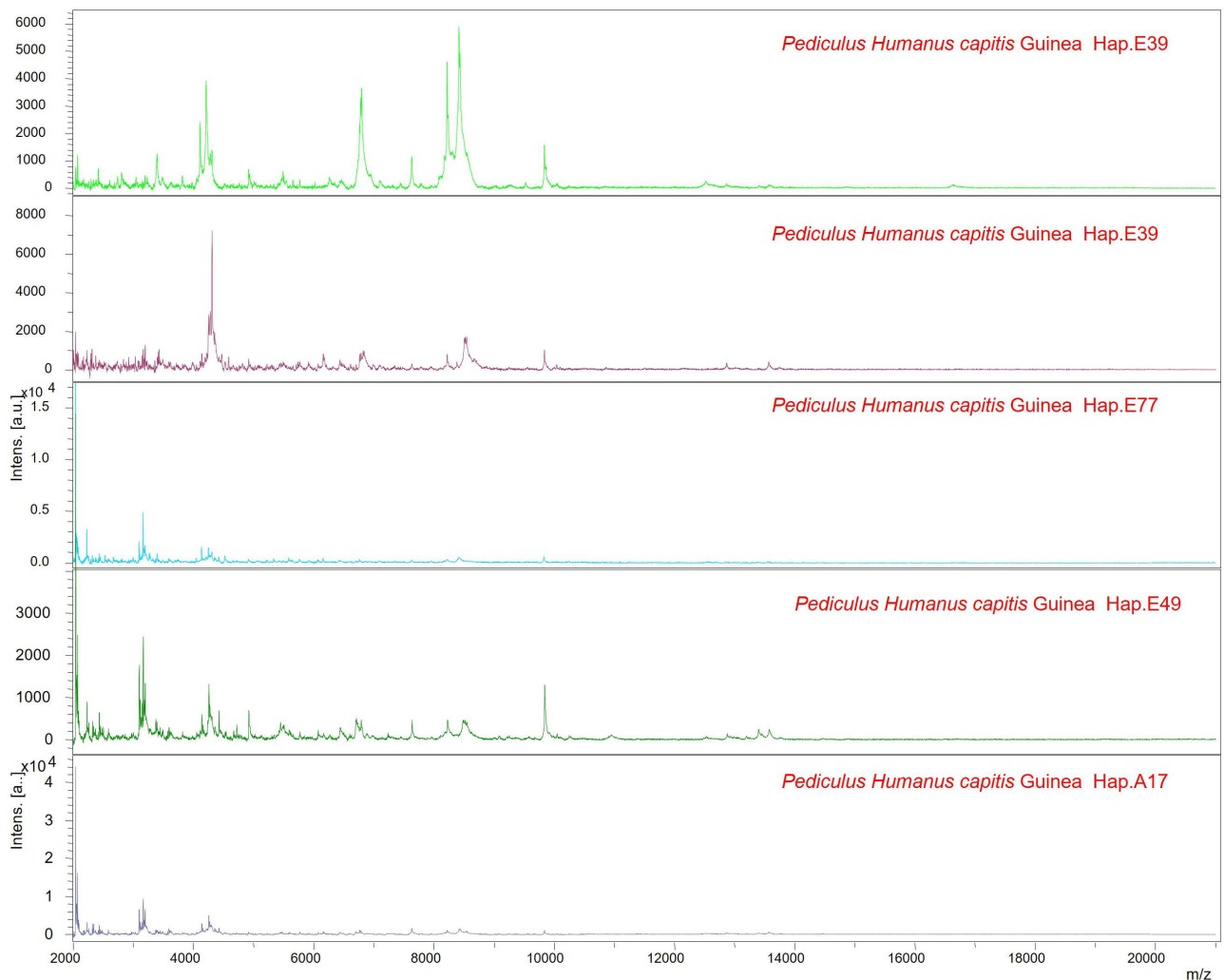

**Fig 3. MALDI-TOF MS spectra representative of the various lice haplotypes.**

DNA isolation. Total DNA was extracted using the Kingfisher Flex System (Thermo Fisher Scientific) with the NucleoMag Pathogen Isolation Kit (MACHEREY-NAGEL GmbH, Düren, Germany) according to the manufacturer's instructions. The extracted DNA was then stored at -20°C until further use.

**2.7.2. Genotypic analysis of lice. 2.7.2.1. Mitochondrial haplogroup identification** To determine the mitochondrial clade of the lice collected in this study, all DNA samples were subjected to clade-specific real-time duplex qPCR analysis targeting a fragment of the cytochrome b (cytb) gene. Each duplex qPCR system is specific for haplogroups A-D and B-C, with the particularity that the B-C duplex also amplifies samples belonging to clade E, which is classified as a subclade of clade C [23]. This qPCR can therefore detect both haplogroups C and E. To differentiate between them, we then performed another monoplex qPCR assay, specific for clade E only, targeting a 129 bp region (nucleotide position 605–734) of the cytb gene [24]. In addition, all lice that tested positive on the B-C/E qPCR system were subjected to a clade E specific qPCR. To ensure the reliability and quality of the assays, lice carrying a previously identified haplogroup were used as positive controls and negative controls were included in each assay. Clade identification by qPCR was performed using the CFX96 real-time system (Bio-Rad Laboratories, Foster City, CA, USA). DNA amplification was

performed with the following parameters: an incubation step at 50°C for 2 min (to activate the UDG), followed by an initial denaturation at 95°C for 5 min, 45 cycles of denaturation at 95°C for 5 s, and a hybridisation step at 60°C for 30 s. The final reaction volume was 20 µL, containing 10 µL of LightCycler 480 Probes Master (Roche, Germany), 0.5 µL of uracil DNA glycosylase, (UDG,1U/µL), 0.5 µM of each primer and probe, and water [23].

**2.7.2.2. Lice haplotype determination:** For haplotype determination and phylogeny analysis, we use standard PCR targeting a 347 bp fragment of the mitochondrial cytb gene with previously described primers [23]. PCR amplification was performed on a MiniAmp thermal cycler (Applied Biosystems by Thermofisher Scientific, France). The final PCR volume was 25 µL, consisting of 12.5 µL Amplitaq Gold Master Mix, 0.5 µm of each primer, 5 µL DNA template and water. The PCR thermal profile included an initial incubation period of 15 minutes at 95°C, followed by 40 cycles of 1 minute at 95°C, 30 seconds at 56°C and 1 minute at 72°C, with a final extension step of 5 minutes at 72°C [23].

Successful amplification of the target sequence was confirmed by agarose gel electrophoresis. The amplified products were prepared and sequenced using methods analogous to those previously described [23]. PCR products were purified using NucleoFast 96 PCR plates (Macherey-Nagel EURL, Hoerdt, France), according to the manufacturer's instructions. The amplicons were then sequenced using the Big Dye Terminator Cycle Sequencing Kit (Perkin Elmer Applied Biosystems, Foster City, CA, USA) on a 3500×L genetic analyser (AB Applied Biosystems, HITACHI, Japan). The electropherograms were assembled and edited using ChromasPro software (ChromasPro 1.7.7, Technelysium Pty Ltd, Tewantin, Australia), and a comparison was made with those available in the GenBank database using NCBI BLAST (http://blast.ncbi.nlm.nih.gov/Blast.cgi).

**2.7.3. Sequences and phylogenetic diversity analysis.** To assess clade diversity in human lice, the total cytochrome b (cytb) sequences of *Pediculus humanus capitis* from Guinea obtained in this study were pooled and compared with previously reported mitochondrial cytb sequences of *Pediculus humanus capitis* from around the world. Alignments were performed using ClustalW software integrated into MEGA version 11.0.13. Maximum likelihood (ML) analysis was also performed in MEGA11 using the Tamura-Nei model for nucleotide sequences with 1000 bootstrap replications [25]. The cytochrome b (cytb) gene from *Pediculus schaeffi* (AY696067) was used as an outgroup. Finally, the Interactive Tree of Life (ITOL) application was used to investigate the potential relationships between haplotypes (itol.embl.de).

**2.7.4. qPCR detection of lice-borne pathogen DNA.** To determine the presence of lice-borne pathogens, DNA was extracted from all lice samples and subjected to quantitative polymerase chain reaction (qPCR) analysis targeting several genes of different pathogenic bacteria. These included the gltA gene from *Rickettsia spp.* and the 16S rRNA gene from *Borrelia spp.*, the yopP gene from *B. quintana*, the PLA gene from *Yersinia pestis*, the IS1111 gene from *Coxiella burnetii*, the 23S rRNA gene from Anaplasmataceae. *Acinetobacter* spp. were also tested. Previously developed specific primers and probes were used (Table 1). The qPCR analysis was conducted according to the previously described protocol for the cytb gene, with the inclusion of a positive target DNA control and a negative master mix control in each PCR run. *Acinetobacter* spp. positive samples then underwent an *Acinetobacter baumannii* specific qPCR directed at the OmpA/MotB gene (Table 1).

## 2.8. Patients and clinical diagnosis of scabies

This study population consisted of schoolchildren suspected of having clinical scabies. Skin scrapings were taken from 20 patients suspected of having scabies, and swabs were taken from the clinically suspected lesions. The typical distribution of these lesions was determined by observing their occurrence on the fingers, wrists, hands, armpits, buttocks and knees.

## 2.9. Scabies DNA extraction and real-time PCR

Swab samples were placed in 2 ml tubes together with ceramic beads (1.4 mm) and 700 µL lysis buffer (NUCLISENS easyMAG, bioMérieux SA, France) for mechanical lysis. Mechanical lysis was performed at a speed of 6 m/s for 40 s

**Table 1. Sequences of primers and probes used for qPCR to detect lice-borne pathogens.**

| Target | Name | Primers and Probes (5'-3') | References |
|---|---|---|---|
| *P. humanus* | *Cytb. Duplex A/D* | FAM-CATTCTTGTCTACGTTCATATTTGGTAMRA | [26] |
| | | VIC-TATTCTTGTCTACGTTCATGTTTGA-TAMRA | |
| | | F_ GATGTAAATAGAGGGTGGTT | |
| | | R_ GAAATTCCTGAAAATCAAAC | |
| | *Cytb. Duplex B/C-E* | FAM-GAGCTGGATAGTGATAAGGTTTAT-TAMRA | |
| | | VIC-CTTGCCGTTTATTTTGTTGGGGTTT-TAMRA | |
| | | F_ TTAGAGCGMTTRTTTACCC | |
| | | R_ AYAAACACACAAAAMCTCCT | |
| | *Cytb* | F_ GAGCGACTGTAATTACTAATC | [27] |
| | | R_ CAACAAAATTATCCGGGTCC | |
| *Rickettsia spp. Citrate synthase (glta)* | RKND03 | FAM-CTATTATGCTTGCGGCTGTCGGTTC-TAMRA | [28] |
| | | F_GTGAATGAAAGATTACACTATTTAT | |
| | | R_GTATCTTAGCAATCATTCTAATAGC | |
| *Borrelia spp. 16 s ribosomal RNA* | Bor 16s | FAM-CCGGCCTGAGAGGGTGAACGG-TAMRA | [29] |
| | | F_AGCCTTTAAAGCTTCGCTTGTAG | |
| | | R_GCCTCCCGTAGGAGTCTGG | |
| Bartonella quintana | Hypothetical intercellular | FAM-GCGCGCGCTTGATAAGCGTG-TAMRA | [30] |
| | | F_GATGCCGGGGAAGGTTTTC | |
| | | R_GCCTGGGAGGACTTGAACCT | |
| *Yersinia pestis* | PLA | FAM-TCCCGAAAGGAGTGCGGGTAATAGG-TAMRA | [31] |
| | | F_ATG GAG CTT ATA CCG GAA AC | |
| | | R_GCG ATA CTG GCC TGC AAG | |
| *Coxiella burnetii* | IS1111 | FAM-CCGAGTTCGAAACAATGAGGGCTG-TAMRA | [32] |
| | | F_ CGCTGACCTACAGAAATATGTCC | |
| | | R_ GGGGTAAGTAAATAATACCTTCTGG | |
| Anaplasma spp. 23s ribosomal RNA | TtAna | FAM-GGATTAGACCCGAAACCAAG-TAMRA | [33] |
| | | F_TGACAGCGTACCTTTTGCAT | |
| | | R_TGGAGGACCGAACCTGTTAC | |
| *Acinetobacter spp. rna polymerase β subunit gene* | rpoB | FAM-CGCGAAGATATCGGTCTSCAAGC-TAMRA | [34] |
| | | F_TACTCATATACCGAAAAGAAACGG | |
| | | R_GGYTTACCAAGRCTATACTCAAC | |
| | *rpoB (zone1)* | FAM-CGCGAAGATATCGGTCTSCAAGC-TAMRA | [35] |
| | | F_TACTCATATACCGAAAAGAAACGG | |
| | | R_GGYTTACCAAGRCTATACTCAAC | |
| | rpoB (zone 1) | F_TAYCGYAAAGAYTTGAAAGAAG | [23] |
| | | R_CMACACCYTTGTTMCCRTGA | |
| | | F_ TACAARATCTTYGAAGAAGC | |
| *Acinetobacter baumanii. type VI secretion system ompa/motb* | ompA /MotB | FAM_AAGTCGCCAAGAAACCTTGA_TAMRA | [36] |
| | | F_TCAACATCACAATCTTTAGTAGCTGA | |
| | | R_CGCTCTTGCCAGCATAAAGA | |
| Scabies | (ITS2) Second Internal transcribed Spacer | FAM_AAAGCACATCGATGGTGCGA_TAMRA<br>F_ CTTTTTGAATGAATTTGCTG<br>R_ ATCTGAGGTCGAGAAATGAC | [37] |

using FastPrep-24 (Thermo Fisher Scientific), followed by centrifugation at 13,000 rpm for two minutes. DNA was then extracted using the Kingfisher Flex system (Thermo Fisher Scientific) with the NucleoMag Pathogen Isolation Kit (MACHEREY-NAGEL GmbH & Co, Düren, Germany) according to the manufacturer's instructions. The extracted DNA was eluted in 100 µL and stored at -20°C. Primers and probes targeting the ITS-2 gene of Sarcoptes scabiei with a fragment length of 400 base pairs (Table 1) were used. The final reaction volume of 20 µL consisted of 10 µL LightCycler 480 Probes PCR Master Mix (Roche, Germany), 1.1 µL of each primer (20 µM), 1 µL probe (5 µM), 1.8 µL water and 5 µL DNA. DNA amplification was performed using the following cycling conditions: initial denaturation at 95°C for 10 minutes, followed by 40 cycles of 15 seconds at 95°C and 45 seconds at 60°C. A positive control containing the target DNA and a negative control of the master mix were added.

## 3. Results

### 3.1. Description of pediculosis

Thirty-three school girls, representing 8% of the study population, presented with head lice. They were aged between 5 and 12 years, with the majority in the 11-year-old group (44%). Notably, no males were found to be infested. Of the total number of girls affected, 27 presented with mobile lice and 28 presented with severe itching. We collected 81 head lice from 18 girls in the Boké and Kankan regions, specifically at Goreye Primary School in Boké, Bordo 1 Primary School in Kankan, and Karifamoriyah Sub-Prefecture at Diankana Primary School.

Two girls were from Boke town, 8 from Bordo and 8 others from Diankana. A total of three head lice were isolated from the two girls in Boke, 41 head lice from the eight girls in Bordo 1 (Kankan) and 37 head lice from the eight girls in Diankana village (Kankan). No body lice were found during the survey.

### 3.2. Head lice processing

Eighty-one lice, preserved in 70% alcohol were pre-treated before analysis. The lice were first rinsed with distilled water and then dried on a blotting paper. Their characteristics were carefully observed with a ZEISS (France) Axio Zoom V16 binocular loupe and documented by photographs taken with a Canon (France) EOS 7D digital camera equipped with a Canon MP-E 65 mm lens. Morphological keys were provided by François-Xavier Pajot [19] were used for morphological identification (Fig 2 before the lice were dissected for further MALDI-TOF MS and DNA analysis.

### 3.3. Morphological identification

In total, seventy-nine (98%) head lice could be identified by their morphological characteristics. Of these, forty-six (58%) were male and thirty-three (42%) were female; sixty-seven (84%) were in the third larval stage (L3), 11% in the second larval stage (L2) and 4% in the first larval stage (L1). Two samples in the batch could not be identified due to advanced deterioration (Fig 2).

### 3.4. MALDI-TOF MS lice identification

A total of 79 lice specimens were subjected to MALDI-TOF MS analysis. High-quality MS spectra, i.e., spectra with an intensity greater than 3000 arbitrary units and minimal background noise, were obtained for 70 of the 79 specimens (87%). For database construction, four MS spectra from five specimens, whose morphological identifications were confirmed by molecular analysis, were used to generate main spectrum profiles (MSPs) and update our MALDI-TOF MS reference database. A representative MS spectrum from each of these five specimens is shown in Fig 3. Blind testing of the MS spectra from the remaining 65 specimens, which were not included in the updated database, correctly identified them all as *Pediculus humanus capitis*, with log score values ranging from 1.708 to 2.339 (mean: 2.05).

Blind testing of the remaining 65 spectra (not included in the MALDI-TOF MS database) against our database identified them all as *Pediculus humanus capitis*, with MALDI Biotyper log scores values ranging from 1.708 to 2.339 with a mean of 2.05.

### 3.5. qPCR-based identification of lice mitochondrial haplogroup

The A-D and B-C/E clade systems were applied to all lice and the results of these analyses showed that all samples were positive for the B-C/E clade. Standard PCR followed by sequencing was also carried out and confirmed that 77 sequences showed the dominance of the E clade, while three samples belonged to the A clade. The 79 sequences obtained were subjected to phylogenetic analysis. They were aligned and combined with other sequences available online for cytb gene haplotype analysis. The features of the phylogenetic tree highlighted the existence of two haplogroups. Seventy-seven samples belonged to clade E and 2 samples, one from Goreye primary school in the Boké region and one from the Kankan region, clustered with lice samples of clade A. Further analysis revealed three clade E haplotypes, including one novel one. The clade A samples belonged to haplotype A17, which is the globally distributed clade that includes body lice.

The most common haplotypes in this study were E39 and E49. Of these, haplotype E39 stands out as the most wide-spread of the l haplotypes in the E clade, accounting for the majority of lice samples in our study. The distribution of lice by haplotype was as follows: 71 (90%) belonged to the previously described haplotype E39 and 3 (4%) to haplotype E49. In addition, 3 (4%) lice belonged to the novel haplotype E77 (PQ177850.1), and, two lice (3%) belonged to haplotype A17 (Fig 4).

### 3.6. Louse-borne pathogens detection by qPCR

The qPCR detection of the louse-borne pathogens *Rickettsia* spp, *B. quintana*, *Borrelia* spp, *Y. pestis*, *C. burnetii* and *Anaplasma* spp. was negative in our samples. However, Acinetobacter spp. DNA was detected in 37 (46%) of the 81 head lice specimens. The *Acinetobacter baumannii* specific qPCR performed on all *Acinetobacter* spp. positive lice, was positive in three (8%) lice. Unfortunately, the 350 bp fragment of the *Acinetobacter* spp. rpoB gene could not be sequenced and analysed for further identification due to low DNA levels.

Lice harbouring *Acinetobacter* spp. were detected at all study sites. It is important to note that A. baumannii species were detected in the lice collected from the Goreye site in Boké and the Diankana site in Kankan. In one schoolgirl, *A. baumannii* DNA was detected in two out of nine lice.

### 3.7. Clinical diagnosis of scabies

We examined 20 suspected scabies lesions in itchy children, 10 girls and 10 boys, aged between 3 and 17 years, distributed over four primary schools. Of these, 11 (55%) were from the Boké region, seven (35%) from Guéckédou, one (5%) from Kankan and one case (5%) from Labé. Severe pruritus was observed in 40% of the cases, mild pruritus in 30% and moderate pruritus in 25%. In 19 cases, the symptoms resembled those of classic scabies and were localised between the fingers, with scabs on the buttocks, back and ankles. Scabies-specific qPCR was negative in all samples.

## 4. Discussion

Our large epidemiological study of schoolchildren, covering all socio-demographic and eco-climatic areas of the country, adds important new data to the scarce epidemiological data on ectoparasites in the Republic of Guinea. We found that the prevalence of head lice infestation was 7%, while 4% of children presented with lesions compatible with scabies. Consistent with our findings, pediculosis remains a major endemic problem in several regions of Africa [38–40]. In Egypt, a study of primary school children showed an infestation rate of approximately 33% by dermoscopy and 15% by visual inspection [41].

Scabies remains common in disadvantaged communities, such as suburban and rural populations, with prevalence ranging from 9% to 19% [42]. The prevalence of clinical scabies is similar to that reported in Egypt (4.4%) and southern Ethiopia (5.5%) [43]. It is lower in Côte d'Ivoire (1%) [44]. However, a high prevalence of clinical scabies was observed in children aged 5–14 years in the Meta Robi district of Ethiopia, where it reached 19.26% This value is similar to that found in a study conducted in Cameroon (17.8%) [45]. The main limitation of these results is that the clinical presentation of

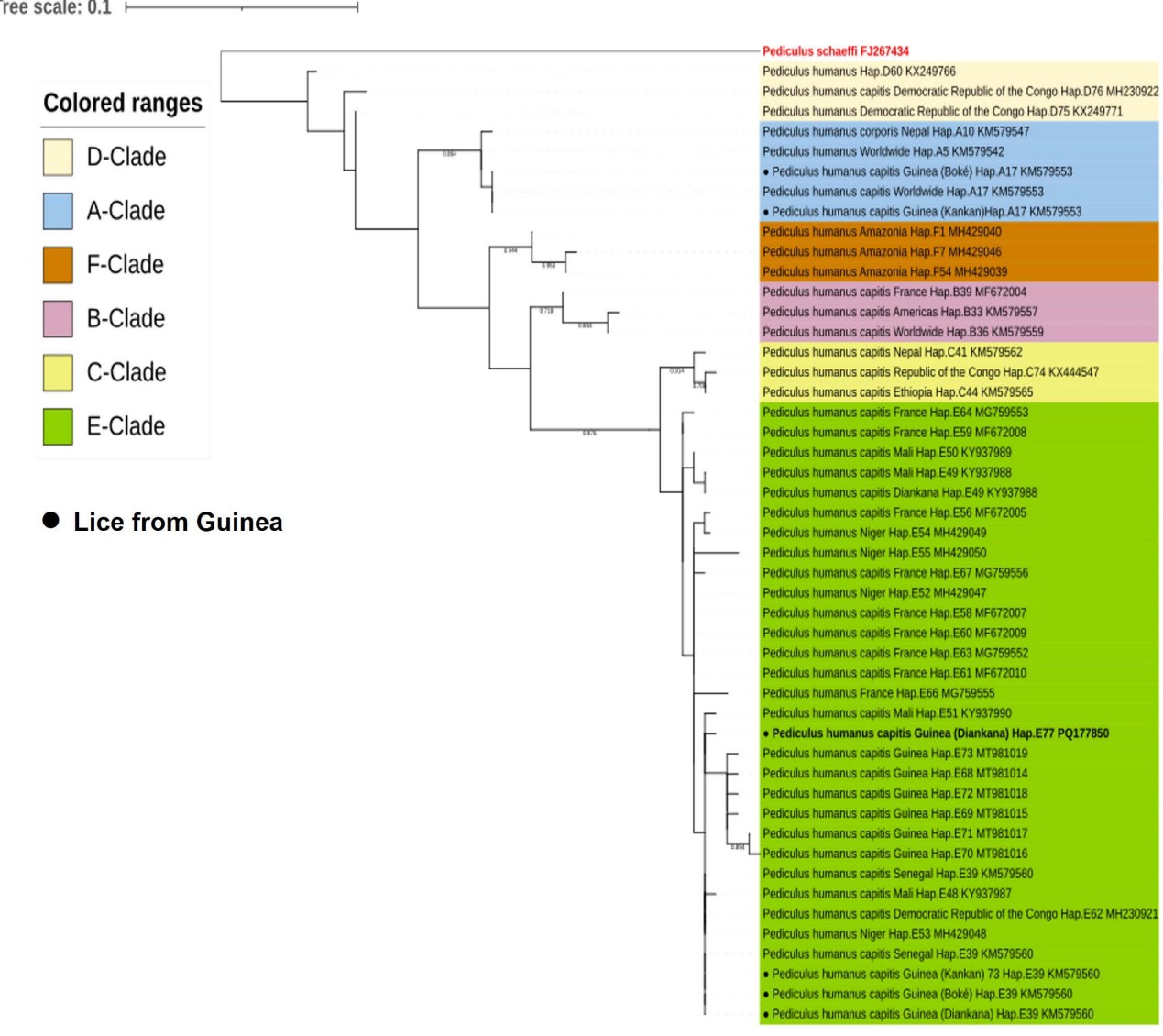

**Fig 4. A maximum likelihood tree of the mitochondrial cytb gene illustrating the relationship between the haplotypes identified in this study and other Pediculus humanus haplotypes previously reported in the literature.** Phylogenetic analysis was performed using maximum likelihood in MEGA 11, applying the Kimura 2-parameter model with 1000 bootstrap replicates. The final data set contained 79 positions.

scabies is heterogeneous and there are many differential diagnoses. This leads to a relatively high risk of a false positive diagnosis of scabies in patients presenting with pruritus.

The 81 lice collected from the 33 children belonged to mitochondrial clade B-C/E. Further investigation using standard PCR followed by sequencing confirmed that all the lice were of haplogroup E, except one which was of clade A. Phylogenetic analysis revealed the presence of seven haplotypes. Three of these were described for the first time in this study. The detection of clade E in both urban and rural communities confirms the endemic dominance of clade E in West Africa, as previously shown [23]. The most common haplotype identified in our collected lice was E39 (97%), followed by E49 (4%). Two head lice collected during this study belonged to haplotype A17, recognised as a global clade including both head and body lice, but reported for the first time in Guinea.

In a study aimed at assessing the presence of lice-borne bacteria in two villages in Mali [26], 100% of the lice analysed belonged to clade E, which is comparable to our study, where 97% of the lice also belonged to clade E, and corresponds to the characteristic West African lice clade distribution. In addition, the presence of different haplotypes, including novel ones, was revealed by phylogenetic analysis in this study, as in ours. A survey of Nigerian refugees and schoolchildren in Algiers revealed a 45% prevalence of clade E, the most abundant in our study, whereas clade A, part of the global haplotype, was found in non-migrant populations [46].

Several studies have demonstrated the presence of DNA from various pathogens transmitted by head and body lice worldwide. The potential pathogenic role of head lice as disease vectors has long been questioned. However, during a trench fever outbreak in Senegal, *Bartonella quintana* was detected in head lice samples. In this study, we performed a qPCR panel to identify lice-borne bacteria. Only *Acinetobacter* spp. were detected in 37 head lice samples. *Acinetobacter baumannii* was identified in 3 of the 37 samples, which is comparable to a previous study conducted in Guinea [23]. Unfortunately, the lack of genetic information prevented us from performing the sequence analysis. One of the lice in our study belonging to clade A, a globally distributed clade that includes body lice, was infected with *Acinetobacter baumannii*. Studies have shown that body lice can also be infected with these bacteria [47,48]. In France, *A. baumannii* was first isolated from body lice samples collected from homeless people. This bacterium has been observed in all lice worldwide [49].

In conclusion, this is the first time that the extent of ectoparasitic infestations such as pediculosis and scabies, as well as the genetic diversity of head lice and their associated pathogens, have been highlighted in schools in the Republic of Guinea. Skin diseases are a serious public health problem in Guinea, particularly in the Boké and Kankan regions where head lice are endemic. This study confirms previous research on the presence of clade E in head lice samples from Guinea and its distribution throughout West Africa.

## Supporting information

**S1 Table. Table detailing the demographic characteristics of their hosts, the haplotype and the *Acinetobacter* spp. or *A. baumannii* infection status of the lice collected in the Republic of Guinea.**
(DOCX)

**S1 File. Checklist.**
(DOCX)

## Acknowledgments

We extend our sincere gratitude to the study participants and their parents or guardians, as well as to the educational authorities of the Ministry of Pre-University Education and Literacy, and the regional, prefectural, and rural authorities. We also express our appreciation to the field team and the village elders for their valuable collaboration and support throughout this study.

Furthermore, we would like to extend our heartfelt thanks to the *Fondation Méditerranée Infection* for its support through the awarding of a scholarship and project funding. Finally, we sincerely acknowledge all individuals who contributed to the success of this project.

## Author contributions

**Conceptualization:** Alpha Kabinet Keita, Stephane Ranque.

**Data curation:** Elisabeth Yawa Diawara.

**Formal analysis:** Elisabeth Yawa Diawara, Adama Zan Diarra, Basma Ouarti.

**Investigation:** Elisabeth Yawa Diawara.

**Methodology:** Alissa Hammoud, Adama Zan Diarra, Basma Ouarti, Stephane Ranque.

**Project administration:** Elisabeth Yawa Diawara, Alpha Kabinet Keita, Stephane Ranque.

**Resources:** Elisabeth Yawa Diawara, Alpha Kabinet Keita.

**Supervision:** Alpha Kabinet Keita, Adama Zan Diarra, Coralie L'Ollivier, Stephane Ranque.

**Validation:** Alissa Hammoud, Basma Ouarti, Coralie L'Ollivier.

**Writing – original draft:** Elisabeth Yawa Diawara.

**Writing – review & editing:** Alpha Kabinet Keita, Alissa Hammoud, Adama Zan Diarra, Basma Ouarti, Coralie L'Ollivier, Stephane Ranque.

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
