## [Decision Letter · Decision Letter 0]

20 May 2025

PGPH-D-25-00741

Ectoparasite survey of schoolchildren in the Republic of Guinea

Dear Dr. %Stephane_Ranque%,

Thank you for submitting your manuscript to PLOS Global Public Health. After careful consideration, we feel that it has merit but does not fully meet PLOS Global Public Health’s publication criteria as it currently stands. Therefore, we invite you to submit a revised version of the manuscript that addresses the points raised during the review process.

We look forward to receiving your revised manuscript.

Kind regards,

Gagandeep Singh, M.D.

Academic Editor

Journal Requirements:

1. Please include a complete copy of PLOS’ questionnaire on inclusivity in global research in your revised manuscript. Our policy for research in this area aims to improve transparency in the reporting of research performed outside of researchers’ own country or community. The policy applies to researchers who have travelled to a different country to conduct research, research with Indigenous populations or their lands, and research on cultural artefacts. The questionnaire can also be requested at the journal’s discretion for any other submissions, even if these conditions are not met.  Please find more information on the policy and a link to download a blank copy of the questionnaire here: https://journals.plos.org/globalpublichealth/s/best-practices-in-research-reporting. Please upload a completed version of your questionnaire as Supporting Information when you resubmit your manuscript. 2. We have amended your Competing Interest statement to comply with journal style. We kindly ask that you double check the statement and let us know if anything is incorrect. 3. We do not publish any copyright or trademark symbols that usually accompany proprietary names, eg (R), (C), or TM  (e.g. next to drug or reagent names). Please remove all instances of trademark/copyright symbols throughout the text, including ®, ™ on page 8, 9, 13. 4. Some material included in your submission may be copyrighted. According to PLOS’s copyright policy, authors who use figures or other material (e.g., graphics, clipart, maps) from another author or copyright holder must demonstrate or obtain permission to publish this material under the Creative Commons Attribution 4.0 International (CC BY 4.0) License used by PLOS journals. Please closely review the details of PLOS’s copyright requirements here: PLOS Licenses and Copyright. If you need to request permissions from a copyright holder, you may use PLOS's Copyright Content Permission form. Please respond directly to this email or email the journal office and provide any known details concerning your material's license terms and permissions required for reuse, even if you have not yet obtained copyright permissions or are unsure of your material's copyright compatibility.  Potential Copyright Issues:a. Figure 1: please (a) provide a direct link to the base layer of the map (i.e., the country or region border shape) and ensure this is also included in the figure legend; and (b) provide a link to the terms of use / license information for the base layer image or shapefile. We cannot publish proprietary or copyrighted maps (e.g. Google Maps, Mapquest) and the terms of use for your map base layer must be compatible with our CC-BY 4.0 license.  Note: if you created the map in a software program like R or ArcGIS, please locate and indicate the source of the basemap shapefile onto which data has been plotted. If your map was obtained from a copyrighted source please amend the figure so that the base map used is from an openly available source. Alternatively, please provide explicit written permission from the copyright holder granting you the right to publish the material under our CC-BY 4.0 license. Please note that the following CC BY licenses are compatible with PLOS license: CC BY 4.0, CC BY 2.0 and CC BY 3.0, meanwhile such licenses as CC BY-ND 3.0 and others are not compatible due to additional restrictions.  If you are unsure whether you can use a map or not, please do reach out and we will be able to help you. The following websites are good examples of where you can source open access or public domain maps: * U.S. Geological Survey (USGS) - All maps are in the public domain. (http://www.usgs.gov) * PlaniGlobe - All maps are published under a Creative Commons license so please cite “PlaniGlobe, http://www.planiglobe.com, CC BY 2.0” in the image credit after the caption. (http://www.planiglobe.com/?lang=enl) * Natural Earth - All maps are public domain. (http://www.naturalearthdata.com/about/terms-of-use/)

Additional Editor Comments (if provided):

Based on the reviews received, I have decided for the manuscript to undergo a 'Major Revision'.

Reviewers' comments:

Reviewer's Responses to Questions

**Comments to the Author**

1. Does this manuscript meet PLOS Global Public Health’s publication criteria?

Reviewer #1: Yes

Reviewer #2: Yes

2. Has the statistical analysis been performed appropriately and rigorously?

Reviewer #1: N/A

Reviewer #2: Yes

3. Have the authors made all data underlying the findings in their manuscript fully available (please refer to the Data Availability Statement at the start of the manuscript PDF file)?

Reviewer #1: Yes

Reviewer #2: Yes

4. Is the manuscript presented in an intelligible fashion and written in standard English?

Reviewer #1: Yes

Reviewer #2: Yes

Reviewer #1: I appreciate the efforts undertaken in this ectoparasite survey. I have a few comments and suggestions that could help further strengthen the manuscript:

1. Abstract: Please mention the creation/expansion of the MALDI-TOF MS database for lice and the fact that scabies PCR was negative in all samples should be mentioned in the abstract, as these are key findings.

2. Methods

i. Please specify how many spectra were collected per specimen for library preparation, as reproducibility of MS spectra is critical.

ii. Clarify what gold standard was used for lice identification for MALDI-TOF library creation. It appears PCR confirmation was used — please explicitly state this in the methods section.

iii. Cluster analysis (MSP dendrogram) could be performed to illustrate how the lice specimens relate to each other, are the halplotypes differentiated on MALDI-TOF?.

iv. In the discussion, kindly mention previous studies where MALDI-TOF MS has been applied for arthropod (especially lice) identification and compare findings where relevant.

v. The stability of cephalothorax MS profiles can be affected by alcohol storage. Ideally, lice should be dried at room temperature for at least 12 hours to avoid alcohol residue affecting MS spectra. You have mentioned drying at room temperature for only 5 minutes, which could be insufficient. Please discuss this as a potential limitation. Suggested reference: Benyahia, H.; Parola, P.; Almeras, L. Evolution of MALDI-TOF MS Profiles from Lice and Fleas Preserved in Alcohol over Time. Insects 2023, 14, 825. https://doi.org/10.3390/insects14100825

3. Results

i. You mention that five lice were identified after library preparation in a blind test. Please include the results of this blind testing

ii. Please clarify what Ct value was considered the cutoff for defining PCR positivity. The Ct values for Acinetobacter spp. detection are very high in supplementary data. It is important to explain the threshold chosen and its implications on detection sensitivity. Furthermore, 37 lice samples were positive for Acinetobacter spp., but A. baumannii PCR was positive in only 3. Possibly, the high Ct cutoff could have led to overestimation.

iii. Please specify how many sequences from your study were used in the maximum likelihood tree of the mitochondrial cytb gene. It appears that only three isolates (one Clade E and 2 clade A) were included. Including more representative sequences would strengthen the phylogenetic analysis. Also all sequences should be submitted in public domain.

4. Discussion

i. Please add a short discussion on other bacteria reported from lice, particularly from the West African region. Suggested reference: Deng, Yuan-Ping et al., Emerging Bacterial Infectious Diseases/Pathogens Vectored by Human Lice, Travel Medicine and Infectious Disease, 2023.

ii. In a recent study from Nigeria, lice were placed mostly into clade A and a few into clade E (Kamani et al., Parasitology Research 2023). Since both studies are from West Africa, it would be valuable to discuss why clade distributions might differ.

Reviewer #2: 1. Could you explain why block randomisation was used in an observational study or stratified randomisation was used, and it is misrepresented as block randomisation?

2. Line 293: Any hypothesis why males were not infested with Lice?

3. Line 259: Kindly introduce Bartonelle quintana as B. quintana before using the short form.

4. Line 346: Before using the short form, kindly introduce Yersinia pestis as Y. pestis.

4. Line 352: Kindly introduce Acinetobacter baumannii as B. quintana A. baumannii before using the short form.

5. After introducing the short form, please italicise the short form. Do not randomly use the complete, abbreviated, and short forms for organisms.

6. Kindly mention and elaborate which clades of Pediculus humanus capitis are having Acinetobacter baumannii.

**Do you want your identity to be public for this peer review?** For information about this choice, including consent withdrawal, please see our Privacy Policy

Reviewer #1: No

Reviewer #2: **Yes: ** Raunak Bir

---

## [Decision Letter · Decision Letter 1]

17 Sep 2025

PGPH-D-25-00741R1

Ectoparasite survey of schoolchildren in the Republic of Guinea

Dear Dr. Ranque,

Thank you for submitting your manuscript to PLOS Global Public Health. After careful consideration, we feel that it has merit but does not fully meet PLOS Global Public Health’s publication criteria as it currently stands. Therefore, we invite you to submit a revised version of the manuscript that addresses the points raised during the review process.

We look forward to receiving your revised manuscript.

Kind regards,

Gagandeep Singh, M.D.

Academic Editor

Journal Requirements:

Additional Editor Comments (if provided):

Reviewer #3:

The use of MALDI for the identification of ticks, mosquito and lice are well standardized. The study is well-designed. I have a couple of queries As mentioned below :

1. Whether there was any database available for the lice identification ?

2. If not then, why the used 5 spectra for database creation,on what basis ?, possibly DNA sequence based analysis, need to get clear clarifications.

3. When they created database,how many spectra of the same sample was captured ? Usualy 12-14 spots spectra captured and best 10 out of 14 spectra is selected for creating database, whether they follow the same.

4. It is not clear, after the database creation,what is the 70 spectra identification results.

Reviewers' comments:

Reviewer's Responses to Questions

**Comments to the Author**

Reviewer #1: All comments have been addressed

Reviewer #2: All comments have been addressed

Reviewer #3: (No Response)

publication criteria?

Reviewer #1: Yes

Reviewer #2: Yes

Reviewer #3: (No Response)

3. Has the statistical analysis been performed appropriately and rigorously?

Reviewer #1: N/A

Reviewer #2: Yes

Reviewer #3: (No Response)

4. Have the authors made all data underlying the findings in their manuscript fully available (please refer to the Data Availability Statement at the start of the manuscript PDF file)?

Reviewer #1: Yes

Reviewer #2: Yes

Reviewer #3: (No Response)

5. Is the manuscript presented in an intelligible fashion and written in standard English?

Reviewer #1: Yes

Reviewer #2: Yes

Reviewer #3: (No Response)

Reviewer #1: (No Response)

Reviewer #2: (No Response)

Reviewer #3: The use of MALDI for the identification of ticks, mosquito and lice are well standardized. The study is well-designed. I have a couple of queries As mentioned below :

1. Whether there was any database available for the lice identification ?

2. If not then, why the used 5 spectra for database creation,on what basis ?, possibly DNA sequence based analysis, need to get clear clarifications.

3. When they created database,how many spectra of the same sample was captured ? Usualy 12-14 spots spectra captured and best 10 out of 14 spectra is selected for creating database, whether they follow the same.

4. It is not clear, after the database creation,what is the 70 spectra identification results.

**Do you want your identity to be public for this peer review?** For information about this choice, including consent withdrawal, please see our Privacy Policy

Reviewer #1: No

Reviewer #2: No

Reviewer #3: No

---

## [Editor Report · Decision Letter 2]

4 Nov 2025

Ectoparasite survey of schoolchildren in the Republic of Guinea

PGPH-D-25-00741R2

Dear Dr Stephane Ranque

We are pleased to inform you that your manuscript 'Ectoparasite survey of schoolchildren in the Republic of Guinea' has been provisionally accepted for publication in PLOS Global Public Health.

Best regards,

Gagandeep Singh, M.D.

Academic Editor
